

**Aircraft observations of water-soluble dicarboxylic acids in the aerosols over**
**China**
Yan-Lin Zhang[1,2,*], Kimitaka Kawamura[1,*], Ping Qing Fu[1,3], Suresh K.R Boreddy[1], Tomomi
Watanabe[1], Shiro Hatakeyama[4,5], Akinori Takami[5], Wei Wang[6,†]
[1]Institute of Low Temperature Science, Hokkaido University, Sapporo 060-0819, Japan
[2]Yale-NUIST Center on Atmospheric Environment, Nanjing University of Information
Science and Technology, Nanjing10044, China
[3]LAPC, Institute of Atmospheric Physics, Chinese Academy of Sciences, Beijing 100029,
China
[4]Institute of Symbiotic Science and Technology, Tokyo University of Agriculture and
Technology, Fuchu, Tokyo 183-8509, Japan
[5]National Institute for Environment Studies, Tsukuba, Ibaraki 305-8506, Japan
[6]Chinese Research Academy of Environmental Sciences, Beijing 100012, China
†deceased
*Correspondence to: Y. -L. Zhang (dryanlinzhang@gmail.com) or K. Kawamura
(kawamura@lowtem.hokudai.ac.jp)
Phone: 81-11-706-5457, fax: 81-11-706-7142





**Abstract**

19         Vertical profiles of low molecular weight dicarboxylic acids, related organic

compounds and SOA tracer compounds in particle phase have not yet been simultaneously
explored in East Asia, although there is growing evidence that aqueous phase oxidation of
volatile organic compounds may be responsible for the elevated organic aerosols (OA) in the
free troposphere. Here, we found consistently good correlation of oxalic acid, the most
abundant organics globally, with its precursors as well as biogenic-derived secondary OA (SOA)
compounds in Chinese tropospheric aerosols by aircraft measurements. Anthropogenically
derived dicarboxylic acids (i.e., $C_5$ and $C_6$ diacids) at high altitudes were 4-20 times higher than
those from surface measurements and even occasionally dominant over oxalic acid at altitude
higher than 2 km, which is in contrast to the predominance of oxalic acid previously reported
globally including the tropospheric and surface aerosols. This indicates an enhancement of
tropospheric SOA formation from anthropogenic precursors. Furthermore, oxalic acid-to-
sulfate ratio maximized at altitude of ~2 km, explaining aqueous-phase SOA production that
was supported by good correlations with predicted liquid water content, organic carbon and
biogenic SOA tracers. These results demonstrate that elevated oxalic acid and related SOA
compounds from both the anthropogenic and biogenic sources may substantially contribute to
tropospheric OA burden over polluted regions of China, implying aerosol-associated climate
effects and intercontinental transport.
**1 Introduction**

38         Low molecular weight (LMW) dicarboxylic acids (diacids), one of the most abundant

organic aerosol (OA) constituents, have been reported in the marine (Kawamura and Sakaguchi,
1999), remote (Kawamura et al., 1996), mountainous, rural, semi-urban, and urban atmosphere
(Ho et al., 2007). LMW diacids play important roles in Earth's climate by directly scattering
sunlight or indirectly by enhancing the ability of OA to act as cloud condensation nuclei (CCN)
(Kumar et al., 2003). They can contribute from a few percent of the water-soluble mass



(Sorooshian et al., 2007b), which could serve as tracers for the atmospheric processing of water-
soluble OA (Ervens et al., 2011), to more than 10% of organic carbon (OC) in the remote marine
atmosphere (Kawamura and Sakaguchi, 1999).
LMW diacids in aerosols may be directly emitted from fossil fuel combustion (e.g.
vehicle exhaust) (Kawamura and Kaplan, 1987), biomass burning (Narukawa et al., 1999) and
cooking emissions (Rogge et al., 1991). They can also be formed by degradation from
unsaturated fatty acids (Kawamura et al., 1996) and cyclic alkenes (Hatakeyama et al., 1987).
Laboratory studies have provided direct experimental evidence that stepwise aqueous oxidation
of relatively longer-chain (n) dicarboxylic acids can yield the corresponding short-chain (n-1)
dicarboxylic acids (Enami et al., 2015). In addition, in-cloud and below-cloud measurements
as well as other field measurements have revealed that aqueous-phase oxidation of volatile
organic compounds (VOCs) and intermediates such as glyoxal (Gly), methylglyoxal (MeGly)
and pyruvic acid (Pyr) in wet aerosols or clouds and the subsequent gas-particle partitioning is
more important pathway for the production of small diacids such as oxalic and malonic acids
(Lim et al., 2013;Carlton et al., 2007;Lim et al., 2005;Carlton et al., 2006;Yu et al.,
2005;Sorooshian et al., 2007b;Sorooshian et al., 2006). Oxalic acid ($C_2$) is the most abundant
diacid with concentrations ranging from a few ng m$^{-3}$ in remote locations (Kawamura et al.,
1996) to hundreds or even up to one thousand ng m$^{-3}$ in urban regions (Ho et al., 2007) and
highly forested regions (Falkovich et al., 2005). Indeed, it has been reported that photochemical
oxidation of isoprene is a predominant formation pathway of oxalic acid (Myriokefalitakis et
al., 2011).
Most of the previous studies of diacids have been conducted at ground surface;
however, only a few aircraft observations that have been conducted over the arctic region
(Talbot et al., 1992), the western North Pacific (Narukawa et al., 1999),  the western rim of the
Pacific Ocean (Kawamura et al., 2003), the coastal marine stratocumulus and cumulus clouds
over USA (Crahan et al., 2004;Sorooshian et al., 2013;Sorooshian et al., 2007a). These aircraft
experiments have revealed that water-soluble diacids may be produced by the photochemical





oxidation of anthropogenic organic compounds in the atmosphere and thus play an important
role in controlling the chemical and physical properties of OA in the troposphere. However, the
evidence of secondary production of LWM diacids (i.e., oxalic acid) in tropospheric aerosols
maybe not enough due to the lack of dataset from simultaneous measurements of their possible
precursors and/or intermediates such as particle-phase glyoxal (Gly), methylglyoxal (MeGly),
pyruvic acid (Pyr) and glyoxylic acid ($\omega C_2$) as well as other secondary organic aerosol (SOA)
compounds from photochemical oxidation of VOCs such as isoprene and monoterpenes in the
aerosols collected from aircraft campaigns.
East Asia is one of the most important source regions of OA (Zhang et al., 2007), and
this is especially true for China due to the rapid urbanization, industrialization and energy
consumption (Huang et al., 2014;Zhang et al., 2015a;Zhang and Cao, 2015). Elevated OA were
found in the free troposphere (FT) from ACE-Asia aircraft observations, exceeding model-
predicted organic aerosols by a factor of 10-100 (Heald et al., 2011;Heald et al., 2005;Henze
and Seinfeld, 2006). There is growing evidence that SOA formation from aqueous phase
processing of VOCs, especially from isoprene may partially explain the missing OA in the
troposphere over East Asia (Heald et al., 2011;Heald et al., 2005;Henze and Seinfeld, 2006),
however, vertical profiles of LWM diacids, related organic compounds (i.e., ketoacids and α-
dicarbonyls) and SOA tracer compounds in particle phase have not yet been simultaneously
analyzed in this region. Additional chemical constraints are urgently required to investigate the
importance of SOA formation pathway to the global/regional budget of OA. Our previous
studies have reported the molecular composition of primary organic aerosols (POA) including
n-alkanes, fatty acids, sugars, and polycyclic aromatic hydrocarbons (PAHs) as well as biogenic
SOA tracers in aerosol particles over China (Wang et al., 2007;Fu et al., 2014). However,
chemical composition and molecular distribution of LMW diacids have never been reported by
aircraft measurements over mainland and coastal China, although only limited studies have
been conducted on their spatial distributions based ground observations (Ho et al., 2007).



In this study, we have collected aerosol samples over coastal to inland China using
aircraft during spring, summer and winter. The samples were analyzed for series of LWM
diacids and related compounds as well as SOA tracer compounds to investigate the vertical
profiles, possible sources and formation pathways of LWM diacids and related organic
compounds in the polluted troposphere over China. In light of these analyses, we identified that
production of oxalic acid and related compounds from anthropogenic and biogenic precursors
is enhanced at high altitudes in the troposphere.
**2 Materials and Methods**
**2.1 Aircraft Campaigns**
Aircraft measurements were carried out over the coastal East China Sea in winter
(12/25/2002−01/06/2003, n=18) and inland China during summer (08/08/2003−09/13/2003,
n=14) and spring (05/19/2004−06/10/2004, n=16) using Yun-12 and Yun-5 airplanes as
described elsewhere(Wang et al., 2007;Fu et al., 2014) . The sampling heights were from 500
m to 3000 m above ground level across many major cities such as Changzhou, Nanjing, Hefei,
Wuhan, Chongqing and Chengdu for inland aerosol samples as well as Wenzhou, Ningbo,
Shanghai, Changzhou, Qingdao and Dalian for coastal aerosol samples. The detailed flight
tracks and flight information are shown in Figure 1 and Table S1 (see Supporting Information).
$PM_{2.5}$ aerosols were collected on pre-heated quartz fiber filter (diameter of 90 mm) using a
medium-volume air sampler (Beijing Geological Instrument Factory, China). Aerosol samples
were collected for the sampling period from ca. 80 min. to two hours. Air was taken via an inlet
installed below the cabin of the aircraft at a flow rate of 78 L/min.
**2.2 Measurement of LMW diacids**
LMW diacids and related organic compounds such as ω-oxocarboxylic acids (or
oxoacids), pyruvic acid and α-dicarbonyls were determined as described elsewhere (Kawamura
and Ikushima, 1993;Ho et al., 2010). Briefly, the sample and blank filters were extracted with
organic free ultrapure water (10 ml ×3) in a glass vial for 10 min. To remove insoluble particles





and filter debris, the extracts were passed through a Pasteur pipette packed with quartz wool.
The combined extracts were concentrated using a rotary evaporator under a vacuum and then
further concentrated using a nitrogen blow-down system. The concentrated extracts were
reacted with 14% $BF_3$/n-butanol to derive carboxyl group to dibutyl ester and oxo group to
dibutoxy acetals.

128        The derivatives were extracted with n-hexane, concentrated to near dryness, dissolved

with n-hexane in 1.5 ml glass vials. The samples were finally determined using a gas
chromatography (GC, HP6980) with a split/splitless injection, fused silica capillary column
(HP-5, 0.2 mm i.d. x 25 m long x 0.52 μm film thickness) and a flame ionization detector.
Identification of the compounds was performed by GC retention times with those of authentic
standards and GC/mass spectrometry analyses. Recoveries of authentic standards spiked to a
pre-heated quartz fiber filter were generally better than 85% for all organic compound identified
except for oxalic acid (78%), pyruvic acid (72%), and methylglyoxal (55%). Duplicate analyses
of filter samples from other sites indicated that analytical errors are smaller than 15%. During
the aircraft campaigns, field blank filters were mounted onto the sampler for seconds without
pumping. Blank and sample filters were placed individually in a clean (pre-combusted at 450
℃ for 6 h) glass jar sealed with a Teflon-lined screw cap, transported to the laboratory, and
stored at -20 ℃ prior to analysis (Wang et al., 2007).

141        Small peaks of oxalic, malonic and glyoxylic acids were found in the field and lab

blanks, but they were smaller than 10% of the real samples. The concentrations reported here
are all corrected for the field blanks but not for the recoveries. The method of SOA tracers
measurement was described elsewhere (Fu et al., 2014) .
**2.3 Measurements of EC and OC**

146        The concentrations of organic carbon (OC) and elemental carbon (EC) were measured

with thermal-optical transmittance method by OC/EC Carbon Aerosol Analyzer (Sunset





Laboratory Inc., USA) following the IMPROVE protocol (Chow et al., 2004). All the analyses
of the filter samples were completed in 2006.

**2.4 Estimation of liquid water content and aerosol acidity**

Liquid water content (LWC) and aerosol acidity were predicted by ISORROPIA II,
which is a thermodynamic equilibrium model with robust and rapid convergence for all aerosol
types (Fountoukis and Nenes, 2007). Measured concentrations of water-soluble inorganic ions,
ambient temperature and relative humility were used as input.

**3 Results and discussion**

**3.1 Concentrations and molecular distributions**

Total concentrations of LWM diacids and related compounds quantified in the inland
aircraft $PM_{2.5}$ (i.e., particulate matter with a diameter smaller than 2.5 μm) are 730±328 and
586±457 ng m$^{-3}$ during summer and spring, respectively, while in the coastal aerosols, the
average concentration is 254±209 ng m$^{-3}$ in winter (Table 1). It is of great interest to note that
the concentrations of diacids in the FT over the Arctic and North Pacific are generally much
lower than those reported at ground levels (Kawamura et al., 1996;Kawamura and Sakaguchi,
1999); however, the concentrations in the troposphere over mainland China are within the range
of or even higher than those reported at ground levels in major cities (Ho et al., 2007;Kawamura
and Ikushima, 1993). Due to the enhanced anthropogenic emissions, concentrations of POA
(i.e., n-alkanes, fatty acids, sugars, lignin and resin products, sterols, PAHs, and phthalic acids)
are higher in winter than those in summer and spring (Wang et al., 2007). In contrast, the
concentrations of diacids in summer and spring are 2-15 times higher than those in winter,
indicating that these compounds are mostly of secondary origin via the oxidation of their
gaseous precursors such as isoprene and α/β-pinene (Kanakidou et al., 2005;Carlton et al.,
2006;Carlton et al., 2007;Carlton et al., 2009;Ervens et al., 2011) as discussed below.



Molecular distributions of diacids in our study are generally characterized by the
predominance of oxalic acid ($C_2$) followed by succinic ($C_4$) and malonic ($C_3$) acids during
spring and winter, being consistent with previous findings obtained in Chinese megacities (Ho
et al., 2007) (Figure 2a). However, in many summer samples collected at height of above 2
km, we found the predominance of glutaric ($C_5$) and adipic ($C_6$) acids (see Figure 2b), which
are the major organic compounds produced by the oxidation of anthropogenic cyclohexene and
methylenecyclohexane (Hamilton et al., 2006;Muller et al., 2007). Such a molecular
distribution has not been reported for the free tropospheric [21,27] and ground level aerosols
(Hatakeyama et al., 1987;Enami et al., 2015). In our measurements, the averaged
concentrations of $C_5$ and $C_6$ in summer are 159±79 ng m$^{-3}$ and 93.9±23.2 ng m$^{-3}$, which are 4-
20 times higher than those in ground measurements in many megacities in China (Ho et al.,
2007), Tokyo (Kawamura and Yasui, 2005) and Los Angeles (Kawamura and Kaplan, 1987).
Such high abundances of $C_5$ and $C_6$ observed in summer imply an important formation pathway
associated with enhanced photochemical oxidation of anthropogenic precursors in the polluted
troposphere over China.
**3.2 $C_3$ ($C_2$) to $C_4$ ratio**
The malonic to succinic acid ($C_3/C_4$) ratios can provide information on source,
formation pathways and photochemical aging of organic aerosols. $C_3/C_4$ ratios in aerosols
derived from vehicular exhausts (i.e., 0.25–0.44, av. 0.35) (Kawamura and Kaplan, 1987) have
lower values than those in ambient aerosols from Tokyo (i.e., 0.56–2.9, av. 1.6) (Kawamura
and Ikushima, 1993) and China's megacities (i.e., 0.6–1.1, average 0.74) (Ho et al., 2007). In
contrast, the ratios are substantially higher for aged aerosols because $C_3$ is more produced by
photochemical processing of $C_4$ (Kawamura and Ikushima, 1993). Actually, higher $C_3/C_4$ ratios
are observed for remote marine aerosols from the North Pacific including tropic (range: 1–11,
av. 3.9) which are subjected to extensive aging during the long-range atmospheric transport
(Kawamura and Sakaguchi, 1999). In this study, $C_3/C_4$ ratios are 0.5±0.4 in summer, 0.9±0.1
in spring (Table 1), and 0.7±0.2 in winter, apparently smaller than those in aerosols affected by



atmospheric aging, but similar to that of urban aerosols in China (i.e., 0.9 in summer and 0.6 in
winter) (Ho et al., 2007).
$C_2/C_4$ ratios (2.0 in summer, 4.6 in spring and 4.6 in winter) in our study are found to
be much smaller than those from ground observations in China during summer (inland cities in
summer: av. 7.1) and winter (coastal cities in winter: av. 7.9). The lower $C_2/C_4$ and $C_3/C_4$ ratios
could be also resulting from degradation of $C_2$ and $C_3$ in high altitudes due to increased solar
radiation. However, no strong correlation ($p > 0.05$) is found between $C_2/C_4$ (and $C_3/C_4$) and
sampling altitude although solar radiation is expected to increase with increasing altitudes.
These results suggest that the degradation of higher homologous diacids (i.e., $C_4$) is not an
important pathway for the production of tropospheric $C_2$ and $C_3$ in China, and primary fossil-
fuel emissions and/or secondary production from other precursors are major formation
pathways of these small diacids (i.e., $C_2$, $C_3$ and $C_4$).
**3.3 Cis/trans ratio**
It has been revealed that maleic acid (M, *cis* configuration) is produced by
photochemical oxidation of anthropogenic aromatic hydrocarbons such as benzene and toluene,
which is predominant over fumaric acid (F, *trans* configuration) (Sempere and Kawamura,
1996). M can be photo-isomerized to its *trans* isomer (*F)* in the atmosphere under solar
radiation. M/F ratios (4.9, 4.5 and 6.8 for summer, spring and winter, respectively) are much
higher in the present study than those reported in marine region (0.1-1.5) (Fu et al., 2013) and
Chinese megacities at ground levels (2.0 and 2.2 for summer and winter, respectively) (Ho et
al., 2007). This indicates that only a small fraction of maleic acid is isomerized to fumaric acid
by photochemical transformation and thus SOA produced from anthropogenic emissions in the
low troposphere in China is mostly fresh without substantially photochemical processing (Cong
et al., 2015).
**3.4 $C_5$ ($C_6$) to $C_9$ diacid ratio**





C$_6$ (or C$_5$) to azelaic acid (C$_9$) ratio (i.e., C$_6$/C$_9$ or C$_5$/C$_9$) is often used as an indicator
of relative contribution from anthropogenic and biogenic sources to OA (Kawamura and Yasui,
2005). C$_6$/C$_9$ and C$_5$/C$_9$ ratios during the summer aircraft campaign are on average 17 and 28,
respectively, which are >15 times higher than those reported at the ground surface from major
Chinese cities (Ho et al., 2007;Wang et al., 2002). This comparison further supports that
anthropogenic sources are an important source of OA in the high altitudes over China. Taken
together with other possible SOA components (e.g., oligomers with MW > 250 Da) produced
during the oxidation of anthropogenic VOCs, our result implies that SOA formation plays an
important role in OA budget in the troposphere especially during summer when atmospheric
oxidation capacity is significantly enhanced. It is important to note that the correlation
coefficient of C$_6$ (or C$_5$) with C$_2$ during summer ($r^2$=0.39 or 0.49) is lower than that obtained in
winter ($r^2$=0.76 or 0.59) and spring ($r^2$=0.95 or 0.96), suggesting that C$_2$ has a different
formation pathway and/or its precursors are mostly from biogenic origins (i.e., isoprene) in
summer.

**3.5 Correlation of diacids with SOA tracers**

In the urban atmosphere, dicarboxylic acids can be emitted as primary particles from
motor exhausts (Kawamura and Kaplan, 1987), biomass burning (Cong et al., 2015;Falkovich
et al., 2005), and cooking emissions (Rogge et al., 1991). However, recent field, laboratory and
model studies have demonstrated that aqueous-phase SOA formation from isoprene or other
precursors photooxidation is a major formation pathway of LMW diacids (Myriokefalitakis et
al., 2011;Carlton et al., 2006;Ervens et al., 2011). Furthermore, significant correlations (r>0.70,
p<0.05) are obtained between C$_2$ and its possible precursors and intermediates such as glyoxylic
acid ($\omega$C$_2$), pyruvic acid (Pyr), glyoxal (Gly) and methylglyoxal (MeGly) in all three campaigns
(Table 2). This demonstrates that C$_2$ is produced from its precursor compounds such as Pyr,
Gly, MeGly and $\omega$C$_2$ through the following formation pathways: (CH$_3$COCOOH, HCO−CHO,
CH$_3$COCHO) → HCO−COOH → HOOC−COOH (Carlton et al., 2009;Carlton et al.,
2007;Carlton et al., 2006). Based on the GEOS-Chem Global 3-D chemical transport model,



Gly may be derived from both biogenic and anthropogenic VOCs whereas MeGly is more
specific to biogenic isoprene (Fu et al., 2008). We found that the correlation coefficient between
Gly and $C_2$ is higher than that between MeGly and $C_2$ for winter and spring samples, suggesting
a substantial contribution of SOA formation from anthropogenic VOCs during these two
seasons.
Similarly, $C_2$ also shows better correlations with both anthropogenic-derived SOA such
as $C_5$ and $C_6$ for winter ($r^2$=0.76 or 0.59) and spring ($r^2$=0.95 or 0.96) samples than summer
samples ($r^2$=0.39 or 0.49), further supporting that anthropogenic VOCs play a more important
role in SOA formation than biogenic VOCs during winter and spring. No significant correlation
($r^2$=0.28, $p$>0.05) is found between EC (i.e., a primary tracer for fossil fuel biomass combustion
(Zhang et al., 2015b)) and $C_2$ in summer aerosols, whereas a good correlation is found in spring.
These results indicate that primary emissions are not major sources of $C_2$ during summer, but
their contribution may be more important in spring.
Oxalic acid shows a strong positive correlation with isoprene-derived SOA tracers such
as 2-methylglyceric acid (2-MGA) and $C_5$-alkene triols (cis-2-methyl-1,3,4-trihydroxy-1-
butene,3-methyl-2,3,4-trihydroxy-1-butene  plus  trans-2-methyl-1,3,4-trihydroxy-1-butene)
(Figure 3), but a very weak correlation ($r^2$ = 0.26, $p$>0.05) with 2-methyltetrols (2-
methylthreitol and 2-methylerythritol). Previous studies have revealed that 2-methyltetrols
could be formed through diepoxy derivatives of isoprene through acid-catalyzed hydrolysis
(Wang et al., 2005), whereas 2-MGA is produced by further oxidation of its intermediates such
as methacrolein and methacrylic acid from isoprene (Claeys et al., 2004;Surratt et al., 2006).
Such a good correlation between 2-MGA and oxalic acid could demonstrate that oxalic acid
has a very close link with the higher-generation products of isoprene, which could serve as
precursors of oxalic acid over China. Oxalic acid also significantly correlates with α/β-pinene-
SOA tracers (i.e., pinonic, pinic, 3-hydroxyglutaric, and 3-methyl-1,2,3-butanetricarboxylic
acid) and β-caryophyllene tracer (β-caryophyllinic acid, see Figure 3) (Jaoui et al., 2013).



Overall, both oxalic acid and SOA tracers are more abundant in summer than in spring,
suggesting that production of these organics is associated with higher oxidation capacity,
emission strength and solar radiation in summer. Based on the consistent good correlations of
oxalic acid with SOA tracers derived from isoprene, monoterpene and β-caryophyllene, we
propose that a large fraction of oxalic acid detected within the atmospheric boundary layer over
China is of secondary origin, i.e., mostly via atmospheric oxidation of gaseous precursors
uplifted aloft from ground surface. This study highlights that oxalic acid should serve as an
important tracer of SOA formation not only on the ground surface but also at high altitudes
within the lower FT. Therefore, the high abundances of LWM diacids ($C_2$-$C_6$) observed in this
study imply an important contribution to the OA budget from SOA production from both
biogenic and anthropogenic precursors emitted from the ground surface to high altitudes over
inland China.
The observed total concentrations of oxalic acid and other LWM diacids identified in
this study (Table 1) show the same magnitude as the levels of SOA (i.e., 299±173 ng m$^{-3}$ in
summer and 257±210 ng m$^{-3}$ in spring) estimated by SOA-tracer methods (Fu et al., 2014). It
should be noted that oxalic acid is not included in the traditional "SOA tracer method"
(Kleindienst et al., 2007) and therefore SOA may be underestimated if SOA is calculated by
this approach. Inclusion of oxalic acid (and also other LWM diacids) as major products from
atmospheric oxidation of biogenic (and also anthropogenic) VOCs may partially reduce the
discrepancy between modeled and observed tropospheric OA during the ACE-Asia campaign
(Heald et al., 2005), although further studies are still required to investigate other SOA
compounds (e.g., oligomeric components) produced from anthropogenic and biogenic VOCs
in the reactions to fully understand the associated formation pathway and mechanism.
**3.6 Vertical profiles of LWM-diacids**
As shown in Figure 4, the highest concentrations of oxalic acid and total diacids are
observed around at 2 km in altitude during summer with a sharp decrease toward 3 km.



However, during spring and winter, their concentrations decrease with increasing altitudes due
to the atmospheric dilution during upward transport. Much clear trends are observed in the
vertical profiles of oxalic acid normalized by anthropogenic tracers such as sulfate ($SO_4^{2-}$),
vanadium, and bulk OC. Interestingly, similar trends are also found in spring and winter
samples, but the trends are weaker, suggesting that a secondary production of oxalic acid is
largely enhanced in summer at higher altitudes of the lower troposphere. Similar vertical pattern
has been also reported for biogenic SOA-tracers such as 2-MGA, 3-HGA and MBTCA (3-
methyl-1,2,3-butanetricarboxylic acid), but not for POA such as biomass burning tracers (e.g.,
levoglucosan), fungal spore tracers (arabitol and mannitol), sucrose, and trehalose (Fu et al.,
2014). These results further demonstrate that oxalic acid is mostly likely produced by secondary
process in the troposphere.
Oxalic acid and related organic species identified in the high altitudes could not be
simply explained by uplifting transport of pre-existing SOA produced on the ground surface
because these SOA compounds relative to anthropogenic tracers such as $SO_4^{2-}$, vanadium, and
OC significantly increased with altitude as stated above. This finding suggests that in-situ SOA
production by the oxidation of VOCs lifted from ground surface substantially contributes to the
observed levels of oxalic acid and related species. Therefore, SOA formation in cloud or wet
aerosol via the oxidation of biogenic and anthropogenic VOCs may increase concentrations of
oxalic acid in the lower FT. There is growing evidence to support of in-cloud formation of
oxalic acid and related SOA. Many studies suggest that oxalic acid is mostly produced via
aqueous-phase oxidation of water-soluble organics such as glyoxal, methylglyoxal, pyruvic
acid and glyoxylic acid, which are oxidation intermediates of various VOCs (Ervens et al.,
2004;Carlton et al., 2006;Ervens et al., 2011). Indeed, a good correlation (Figure 5) was found
between predicted liquid water content with both OC and oxalic acid, indicating an important
contribution from SOA formation via cloud processing and/or aqueous-phase oxidation. This
formation pathway is also supported by the consistently good correlation among these species
identified in our study as discussed previously. Aqueous-phase production of oxalic acid and



related compounds may increase the abundances of SOA at the lower FT (around 2 km in this
study). With the GEOS-Chem model based on the Fu et al. (2008) scheme (Fu et al., 2008),
aqueous-phase SOA has a pronounced enhancement in the lower FT (2-6 km) (Heald et al.,
2011), which may explain to some extent the elevated levels of oxalic acid around at ~2 km in
altitude.
**4 Conclusions**
Based on three aircraft measurements over East Asia, this study demonstrates an
aqueous-phase mechanism for SOA productions of diacids in the troposphere following
correlation analysis of oxalic acid in tropospheric aerosols with other measured chemical
variables including its precursors and its intermediate as well as biogenic-derived SOA from
isoprene, monoterpenes and β-caryophyllene. In addition to biogenic-derived SOA compounds,
anthropogenic-derived dicarboxylic acids (e.g. C5 and C6 diacids) are 4-20 times higher than
those from ground measurements and even occasionally dominant over oxalic acid at altitudes
higher than 2 km in summer, which is in contrast to the predominance of oxalic acid previously
reported globally including the tropospheric and surface aerosols. The results suggest an
important formation pathway associated with enhanced photochemical oxidation of
anthropogenic precursors in the polluted troposphere over China. Their relative contribution of
anthropogenic and biogenic sources is subject to future studies. The combination of radiocarbon
($^{14}$C) measurement of water-soluble organic carbon (WSOC) and specific SOA compounds
(e.g., oxalic acid) may provide better insights on biogenic and fossil sources of SOA (Zhang et
al., 2014;Zhang et al., 2015a;Noziere et al., 2015) .
The present study demonstrates that secondary formation of oxalic acid in aqueous
phase plays an important role in the SOA budget from the near surface to the lower FT (i.e., 2
km) over inland China, whereas dilution of pre-existing particles and VOCs, photochemical
decomposition and aerosol processing may decrease the levels of oxalic acid and related
compounds at higher altitudes (>2 km). Our findings also highlight that water-soluble LMW



diacids and other SOA components may control the chemical compositions, physical properties
and budget of OA in the polluted troposphere over China, and thus significantly affect the
regional/global climate and intercontinental transport especially over the Pacific Ocean.

**Acknowledgements**
The data reported in the paper are presented in the Supplementary Materials or are
available upon request from the Y.-L.Z. The authors acknowledge the Ministry of Education,
Culture, Sports, Science and Technology for financial support to perform the aircraft campaigns
through Scientific Research on Priority Areas on Atmospheric Environmental Impacts of
Aerosols in East Asia (no. 416, 2002–2005). The authors also acknowledge Dr. Hong Li from
Chinese Research Academy of Environmental Sciences for her support and help during the
aircraft measurements. This study was also supported by a grant-in-aid no. 14204055 and
24221001 from the Japan Society for the Promotion of Science (JSPS).

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



**Table 1.** Average concentrations (ng m$^{-3}$) and concentration ranges (ng m$^{-3}$) for straight chain diacids (C$_2$–C$_{11}$), branched chain diacids (iC$_4$-iC$_6$), unsaturated diacids (M, F and mM), multifunctional diacids (hC$_4$, kC$_3$ and kC$_7$), oxoacids (ωC$_2$-ωC$_4$, ωC$_9$, and Pyr), and α-dicarbonyls (Gly and MeGly) of aerosol samples collected by aircraft campaigns over China.

| | 2003 Summer (inland China) | | 2004 Spring (inland China) | | 2002/2003 Winter (coastal China) | |
|---|---|---|---|---|---|---|
| | Range | Mean (SD$^a$) | Range | Mean (SD) | Range | Mean (SD) |
| Oxalic, C$_2$ | 36.4-401 | 183 (111) | 76.5-918 | 286 (216) | 13.3-425 | 92.6 (94.5) |
| Malonic, C$_3$ | 6.3-131 | 54 (35) | 12.2-216 | 57.5 (47.6) | 1.4-79 | 15 (17.9) |
| Succinic, C$_4$ | 9.4-277 | 117 (68.2) | 16-319 | 69.1 (69.9) | 2.3-87.9 | 21.4 (19.8) |
| Glutaric, C$_5$ | 20.7-289 | 159 (79.1) | 6.5-73.8 | 18.1 (16.3) | 0.9-26.1 | 9.7 (7.3) |
| Adipic, C$_6$ | 52.3-135 | 93.9 (23.2) | 5.7-69 | 19.4 (18.1) | 3.7-34.5 | 13.4 (8.1) |
| Pimelic, C$_7$ | 0-2.5 | 0.8 (0.7) | n.d.-8.1 | 1.7 (2) | 0.3-7.5 | 1.9 (1.7) |
| Suberic, C$_8$ | n.d.$^b$ | n.d. | n.d.-2.7 | 0.2 (0.7) | n.d.-11 | 3.2 (2.9) |
| Azelaic, C$_9$ | 2-13.5 | 5.6 (3) | 2.4-18.2 | 6.3 (4.3) | 2.9-20.6 | 8.5 (4.8) |
| Sebacic, C$_{10}$ | 0.3-3.6 | 1.3 (1.0) | n.d.-8.4 | 3.6 (2.5) | 0-6.9 | 1.1 (1.8) |
| Undecanedioic , C$_{11}$ | 0.8-4.3 | 2.1 (0.9) | n.d.-4.3 | 1.5 (1.3) | n.d. | n.d. |
| Methylmalonic, iC$_4$ | 1.2-5.9 | 3.7 (1.7) | 1.1-12.5 | 4.9 (3) | 0-3.9 | 0.9 (0.9) |
| Methylsuccinic, iC$_5$ | 0.6-10.8 | 4.4 (3.1) | 1.3-27.5 | 5.9 (6.1) | 0.7-23.2 | 5.9 (5.8) |
| Methylglutaric, iC$_6$ | n.d.-1.3 | 0.4 (0.4) | 0.4-5.9 | 1.2 (1.3) | n.d.-2.8 | 0.7 (0.8) |
| Maleic, M | 1.8-12.5 | 6.5 (3.6) | 3.3-22.3 | 9.4 (5) | 1.6-11.1 | 5.7 (2.9) |
| Fumaric, F | 0.1-3.9 | 1.7 (1.1) | 0.5-8.4 | 3.0 (2.1) | 0.1-6.2 | 1.5 (1.5) |
| Methylmaleic, mM | 2.3-15.1 | 6.3 (3.6) | 2.2-18.0 | 7.4 (3.8) | 1.3-8.2 | 4 (2) |
| Hydroxysuccinic, hC$_4$ | 1.7-12.5 | 5.3 (3.4) | n.d.-9.3 | 1.9 (2.2) | n.d.-13.7 | 1.9 (3.2) |
| Ketomalonic, kC$_3$ | 0.4-9.2 | 4.2 (2.5) | n.d.-22.8 | 5.6 (5.3) | n.d.-26 | 5.1 (5.6) |
| Ketopimelic, kC$_7$ | 0.4-8.2 | 3.0 (2.3) | n.d.-18.7 | 4.0 (4.5) | n.d.-3.9 | 0.6 (0.9) |
| Total diacids | 139-1230 | 653 (290) | 148-1780 | 507 (402) | 40.4-757 | 193 (164) |
| n.s. C$_2$-C$_{11}$ | 128-1160 | 615(272) | 128-1630 | 464 (371) | 31.3-678 | 167 (149) |
| Pyruvic acid, Pyr | n.d.-9.6 | 2.9 (3.0) | 0.1-11.4 | 2.1 (2.8) | 0.7-36.5 | 10 (8.6) |
| Glyoxylic, ωC$_2$ | 8.1-89.6 | 37.7 (25.2) | 8.3-146 | 46.0 (38.0) | 6.7-129 | 30.6 (28.9) |
| 3-oxopropanoic, ωC$_3$ | 0.1-9.7 | 3.3 (2.4) | 0.1-1.1 | 0.5 (0.3) | n.d.-1.9 | 0.5 (0.5) |
| 4-oxobutanoic, ωC$_4$ | 0-23.1 | 8.0 (7.0) | 6.8-38.9 | 14.9 (8.5) | 0.6-35.5 | 7.5 (8.6) |
| 9-oxononoic, ωC$_9$ | 3.4-36.2 | 11.6 (8.9) | 0.3-20.5 | 5.8 (5.2) | 0.2-5.5 | 1.8 (1.4) |
| Total Keto acid | 18.7-131 | 63.5 (36.0) | 23.7-178 | 69.3 (48.5) | 12.2-176 | 50.0 (36.9) |
| Glyoxal, Gly | 0.7-14.8 | 4.0 (3.6) | 0.2-9.5 | 2.3 (2.5) | 0.6-23 | 4.3 (5.1) |
| Methylglyoxal, MeGly | 0.6-28.2 | 10.8 (7.7) | 0.8-27.3 | 7.4 (8.3) | 2.5-24.3 | 7.6 (5.6) |
| Total dicabonyls | 1.3-42.9 | 14.8 (11.1) | 1.7-36.8 | 9.8 (10.7) | 3.1-47.3 | 11.8 (10.5) |
| Total | 170-1390 | 731 (329) | 174-1990 | 586 (457) | 68.5-980 | 255 (209) |
| Gly/MeGly | 0.1-1.2 | 0.4 (0.2) | 0.04-1.0 | 0.4 (0.3) | 0.2-0.9 | 0.5 (0.2) |
| M/F | 1.7-13.9 | 4.9 (2.9) | 1.1-13.8 | 4.5 (3.1) | 1.6-27.4 | 6.8 (6.2) |
| ωC$_2$/C$_2$ | 0.1-0.2 | 0.2 (0.03) | 0.1-0.3 | 0.2 (0.04) | 0.2-0.8 | 0.4 (0.1) |
| C$_2$/C$_4$ | 0.6-6.7 | 2.0 (1.6) | 2.9-5.9 | 4.6 (0.9) | 2.8-13.5 | 4.6 (2.3) |
| C$_3$/C$_4$ | 0.2-1.9 | 0.5 (0.42) | 0.7-1.1 | 0.9 (0.1) | 0.4-1.7 | 0.7 (0.3) |
| C$_5$/C$_9$ | 5.2-64.6 | 31.3 (15.6) | 1.6-4.3 | 2.8 (0.7) | 0.1-2.7 | 1.1 (0.6) |
| C$_6$/C$_9$ | 10.0-41.0 | 19.8 (7.21) | 1.3-4.6 | 2.8 (0.9) | 0.5-2.8 | 1.6 (0.6) |

$^a$SD denotes standard deviation (1σ ); $^b$n.d. denotes not detected.




**Table 2.** Correlation coefficients ($r^2$) among oxalic acid ($C_2$), pyruvic acid (Pyr), glyoxylic acid
($\omega C2$), glyoxal (Gly) and methylglyoxal (MeGly) detected in aerosol samples from aircraft
campaigns during summer 2003, spring 2004 and winter 2002/2003 over China. See Table 1
for abbreviations.

| Summer 2003 | | | | | |
|---|---|---|---|---|---|
| | $C_2$ | Pyr | $\omega C_2$ | Gly | MeGly |
| $C_2$ | 1.00 | | | | |
| Pyr | 0.89 | 1.00 | | | |
| $\omega C_2$ | 0.97 | 0.95 | 1.00 | | |
| Gly | 0.75 | 0.86 | 0.85 | 1.00 | |
| MeGly | 0.91 | 0.97 | 0.97 | 0.90 | 1.00 |

| Spring 2004 | | | | | |
|---|---|---|---|---|---|
| | $C_2$ | Pyr | $\omega C_2$ | Gly | MeGly |
| $C_2$ | 1.00 | | | | |
| Pyr | 0.95 | 1.00 | | | |
| $\omega C_2$ | 0.97 | 0.93 | 1.00 | | |
| Gly | 0.96 | 0.95 | 0.97 | 1.00 | |
| MeGly | 0.93 | 0.92 | 0.94 | 0.93 | 1.00 |

| Winter 2002/2003 | | | | | |
|---|---|---|---|---|---|
| | $C_2$ | Pyr | $\omega C_2$ | Gly | MeGly |
| $C_2$ | 1.00 | | | | |
| Pyr | 0.70 | 1.00 | | | |
| $\omega C_2$ | 0.98 | 0.70 | 1.00 | | |
| Gly | 0.92 | 0.69 | 0.90 | 1.00 | |
| MeGly | 0.85 | 0.63 | 0.83 | 0.94 | 1.00 |




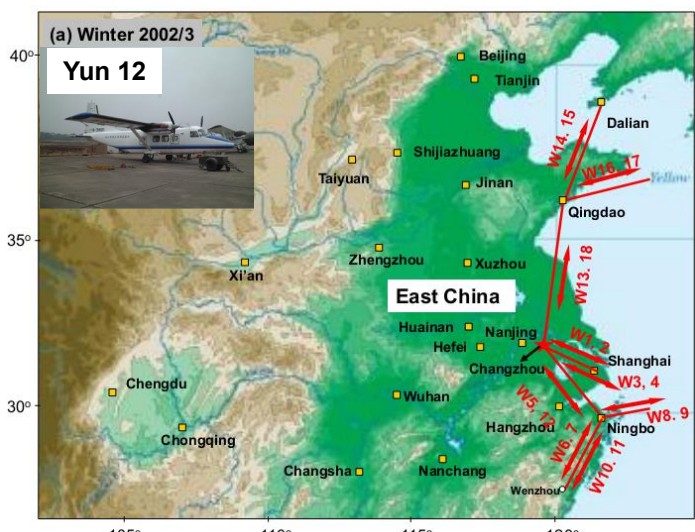


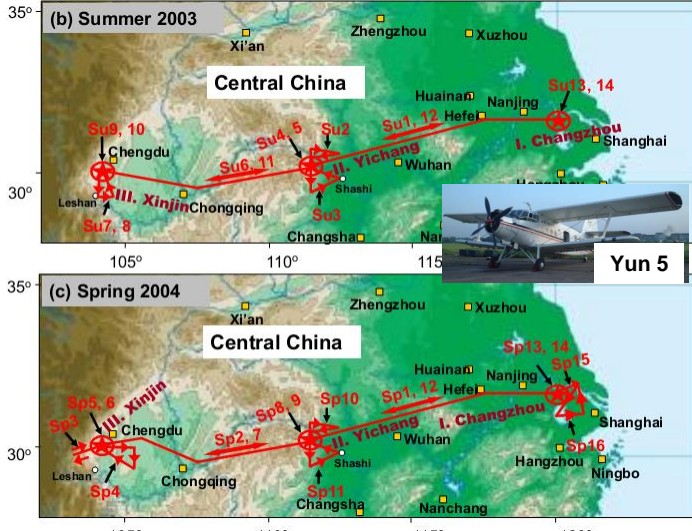


**Figure 1**. Tracks of research flights during aircraft measurements over China during (a) winter
2002, (b) summer 2003 and (c) spring 2004. The detailed sampling information with flight No.
is listed in the Supplementary Table S1. The maps with the flight tracks were drawn by the
software of PowerPoint 2010, https://products.office.com/.



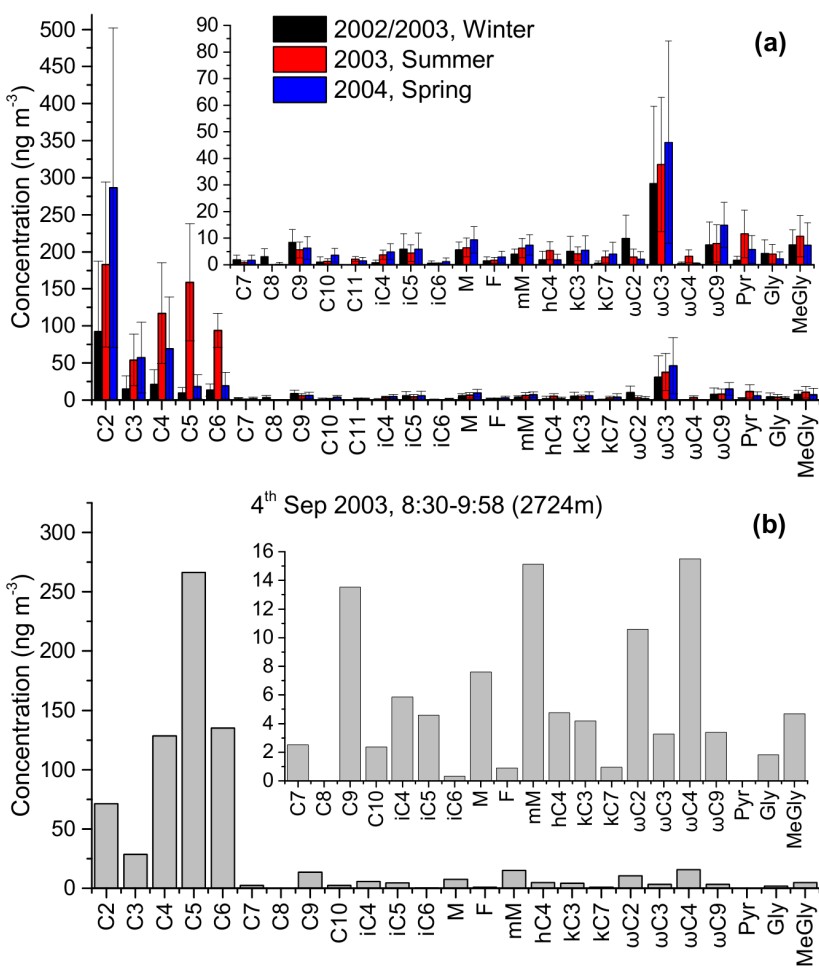


**Figure 2.** Molecular distributions of straight chain diacids ($C_2$–$C_{11}$), branched chain diacids ($iC_4$-$iC_6$), unsaturated diacids (M, F and mM), multifunctional diacids ($hC_4$, $kC_3$ and $kC_7$), ketoacids ($\omega C_2$-$\omega C_4$, $\omega C_9$, and Pyr), and α-dicarbonyls (Gly and MeGly) in aircraft measurement over China; (a) averaged concentrations with bars of standard deviation during winter 2002/2003, summer 2003 and spring 2004; (b) molecular distributions of the measured compounds in the sample collected on $4^{th}$ Sep 2003. See Table 1 for abbreviations.



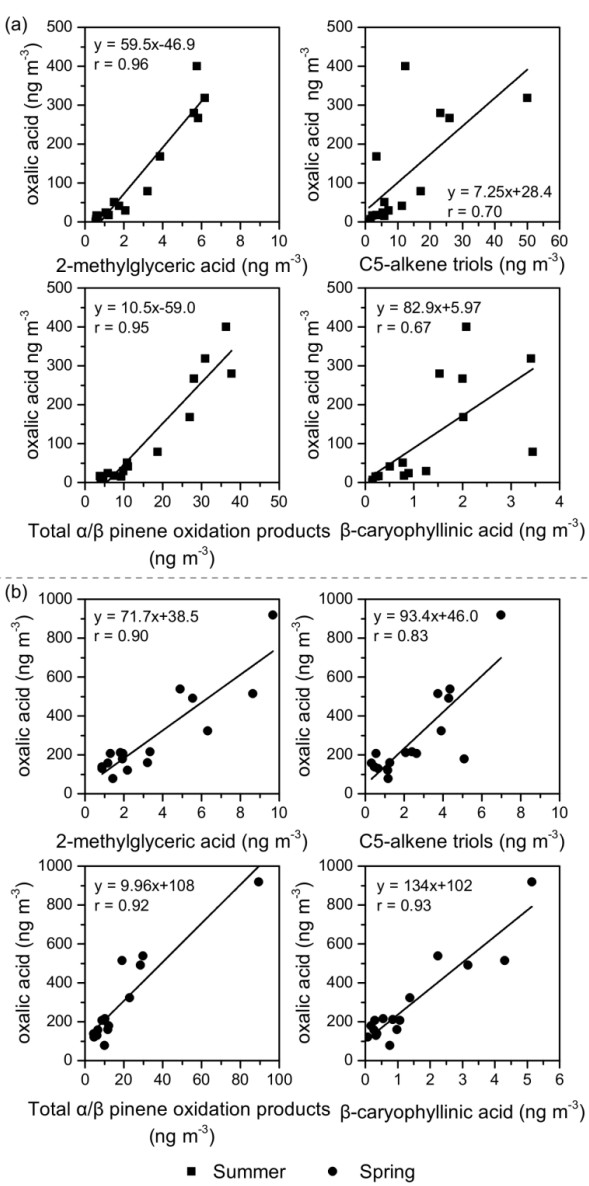

579        ■ Summer    ● Spring

**Figure 3**. Linear relationship of the concentrations of oxalic acid with the detected tracer
compounds for secondary organic aerosols (SOA) such as isoprene SOA tracers such as 2-
methylglyceric acid and C5-alkene triols (sub-total of cis-2-methyl-1,3,4-trihydroxy-1-butene,
3-methyl-2,3,4-trihydroxy-1-butene, trans-2-methyl-1,3,4-trihydroxy-1-butene), α/β-pinene
SOA tracers (subtotal of 3-hydroxyglutaric acid, pinonic acid, pinic acid, 3-methyl-1,2,3-
butanetricarboxylic acid), and β-caryophyllene SOA tracer (i.e., β-caryophyllinic acid) from
aircraft measurements over China during (a) summer and (b) spring.






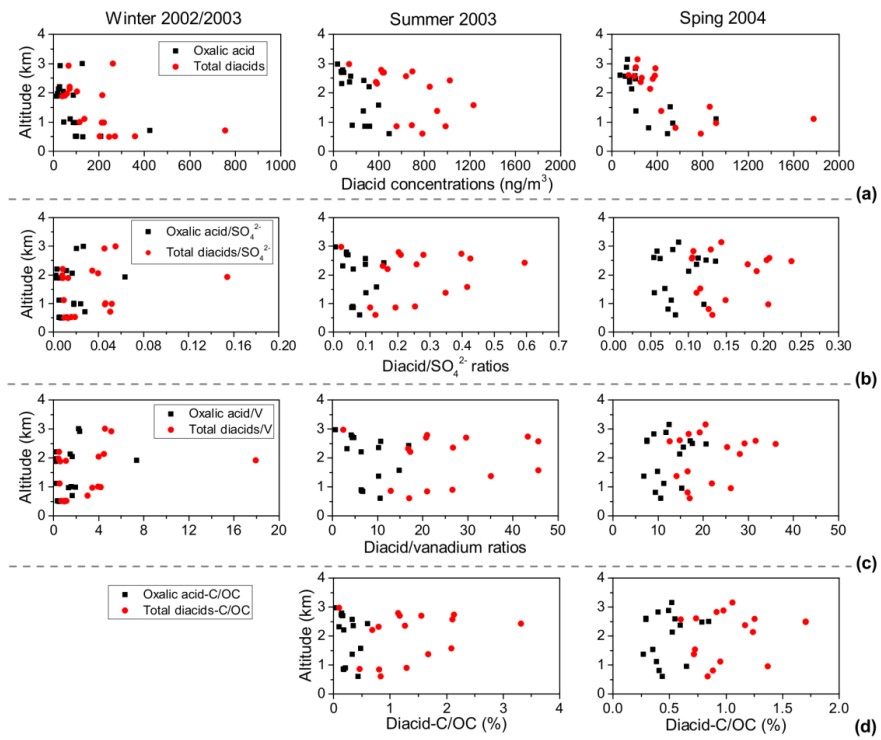


**Figure 4.** Vertical profiles of (a) concentrations of oxalic acid and total diacids, and their relative abundance to (b) sulfate ($SO_4^{2-}$), (c) vanadium (V), and (d) organic carbon (OC, %) in aerosol samples collected during winter 2002 (left), summer 2003 (middle) and spring 2004 (right) aircraft campaigns over China.



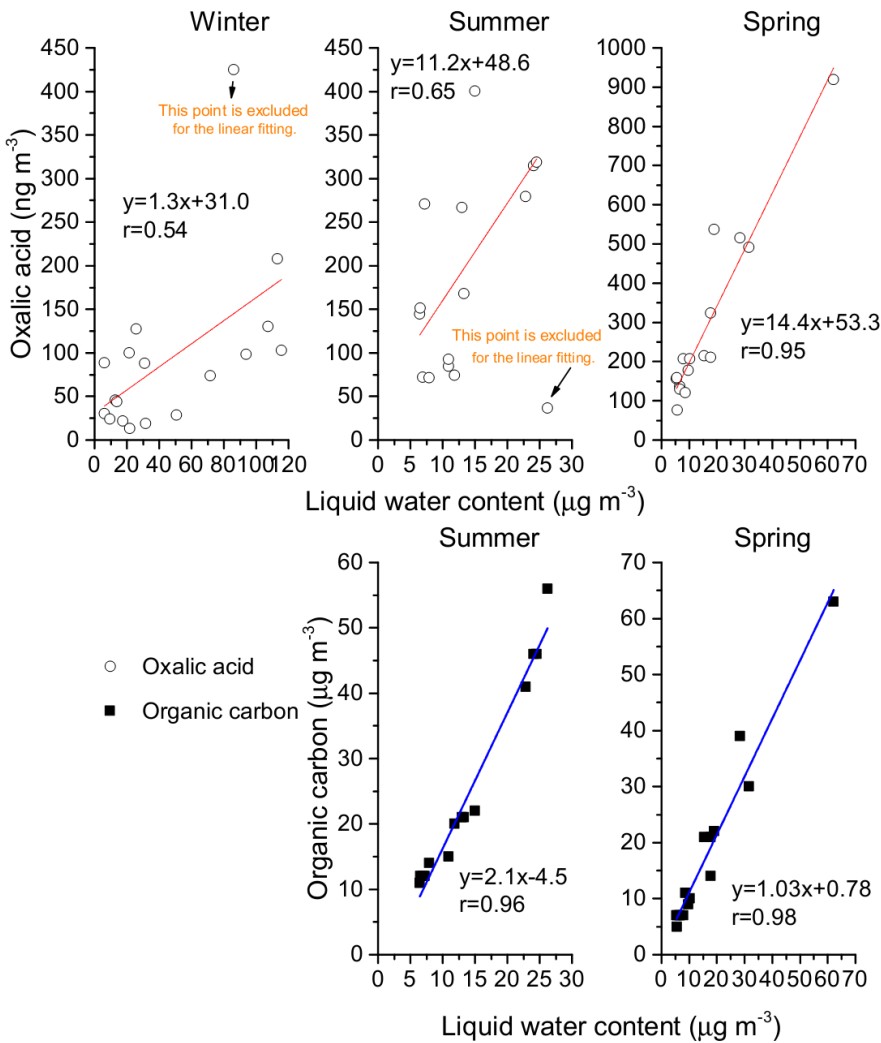

593

**Figure 5**. The linear fitting of the predicted liquid water content with organic carbon (OC) and
oxalic acid aerosols during winter, summer, and spring aircraft measurements over China.