# Peer review of "Aircraft observations of water-soluble dicarboxylic acids in the aerosols over"

_Atmospheric Chemistry and Physics, 2016_

## Referee Comment (RC1) · Anonymous Referee #1 · 15 Feb 2016

The authors present a summary of airborne filter measurement data with a focus on dicarboxylic acids in aerosols over China. The topic of organic acid composition in aerosols is of interest to the atmospheric chemistry community since the organic fraction of particles is complex and uncertain, and requires improvements to be able to model their impacts better. The paper is written fairly well but requires some minor English editing (I point out a few suggested changes below for improving the language). The tables and figures are appropriate. The title and abstract are also appropriate in terms of representing the contents of the manuscript.

The methods used are fine, and the results are informative and rich. The strength of the work includes the detection of so many organic acids that exceed the range of acids detected in some past studies relying on airborne measurements. The overall conclusions mostly repeat those in other studies though, which I hope the authors

can address in their revisions. This study confirms that organic acids are generated by secondary production mechanisms and that aqueous processing can be important, especially aloft. To strengthen the paper, I would suggest that the authors try to find something new in their data to push the state of understanding of organic acids forward as compared to repeating what has already been documented.

The data generated are of high value as airborne diacid data are scarce, so it would be useful to have this work published. But before that happens, as noted above, it is requested that the authors narrow in better on what is novel in their dataset to highlight better in their text. This may be assisted by better consideration of what past work has shown and how this dataset can extend upon those past papers, some of which are highlighted below.

Major Comments:

The authors should expand discussion on the potential influence of biomass burning on their measurements. When was biomass burning evident and how did this impact the organic acid data? Presumably the authors have reasonable tracers for biomass burning plumes.

In various places in the manuscript the authors refer to Free Troposphere (e.g., Line 285). They should make it clear what data and criteria they used to distinguish between FT and the lower mixing layer.

Since denuders are not mentioned in the instrument description, the authors should comment on what impact the lack of using denuders have on the data.

The authors should make note of what affect temperature effects would have on their data as during aircraft sampling there likely are differences in temperature between ambient air and their filters.

Specific Comments: Page 1, Line 4: change 'organics' to 'organic'

Page 2, Line 38-41: Diacids have also been measured in other areas such as deserts

and this should be noted for the sake of completeness (example provided here): Sorooshian, A., et al. (2012). Hygroscopic and chemical properties of aerosols collected near a copper smelter: Implications for public and environmental health, Environ. Sci. Technol., 46, 9473-9480.

Page 2, Line 41: 'play an important role in. . .'

Page 3, Line 66-69: For completeness the authors should refer to the following other airborne studies: For 'Coastal marine stratocumulus and cumulus clouds over USA', add: Wonaschuetz, A., et al. (2012). Aerosol and gas re-distribution by shallow cumulus clouds: an investigation using airborne measurements, J. Geophys. Res., 117, D17202, doi:10.1029/2012JD018089.

Prabhakar, G., et al. (2014). Sources of nitrate in stratocumulus cloud water: Airborne measurements during the 2011 E-PEACE and 2013 NiCE studies, Atmos. Environ., 97, 166-173, doi:10.1016/j.atmosenv.2014.08.019.

Measurements have been conducted over inland agricultural and urban areas in the western United States: Sorooshian, A., et al. (2015). Surface and airborne measurements of organosulfur and methanesulfonate over the western United States and coastal areas, J. Geophys. Res., 120, doi:10.1002/2015JD023822.

Line 98: 'for a series. . .'

Line 325-327: The discussion here can benefit from inclusion of past work showing how the relative amount of oxalate (versus total organic mass) increases with relative humidity based on airborne measurements [Sorooshian, A., et al. (2010). Constraining the contribution of organic acids and AMS m/z 44 to the organic aerosol budget: On the importance of meteorology, aerosol hygroscopicity, and region, Geophys. Res. Lett., 37, L21807, doi:10.1029/2010GL044951] and how water-soluble organics have been shown to be enhanced relative to the surface when humid conditions aloft [Duong, H. T., et al. (2011). Water-soluble organic aerosol in the Los Angeles Basin and outflow

regions: Airborne and ground measurements during the 2010 CalNex field campaign, J. Geophys. Res., 116, D00V04, doi:10.1029/2011JD016674.].

Supplement Table 1: For altitude, the authors should include a plus/minus standard deviation since it seems unlikely that the aircraft was level for that long of a period for each sample.

Line 356: What do the authors mean by the word 'control'? This seems like a very strong word that may not be warranted here. Are these diacids the majority of the OA mass and, if so, can this be shown in the manuscript in a revision?

Table 2: The authors should make mention of the sample number involved with these correlation calculations somewhere in the table or caption.

---

## Referee Comment (RC2) · Anonymous Referee #2 · 29 Feb 2016

The manuscript by Zhang et al studied distributions of organic compounds obtained by aircraft measurements in China and argued that contribution of aqueous-phase photochemistry to SOA formation is significant using various correlation analyses. Potentially this work will make an important contribution to SOA studies particularly in East Asia and be suitable for ACP readership. But some of authors' arguments are confusing, misleading, and not clearly supported by evidence. Most of all, the organization is not effective. I highly recommend that authors should work on organization by bringing the LWC-oxalic acid analysis (currently in the end) to the front in Results and Discussion (Section 3) so that readers are clear about the main point of this manuscript is aqueous chemistry contribution to SOA. This will minimize confusions from paragraphs (Line 279-283, Line 283-285 and Line 291-299). Then, authors can discuss biogenic/anthropogenic influences on oxalic acid formation and aqueous

chemistry. Provide biogenic/anthropogenic emission data if available. Provide NOx measurements, too for isoprene arguments. Take an advantage of referencing most recent papers. Below I provide comments. There are also numerous technical errors (e.g., grammatical errors, typos), which need to be corrected.

Line 38: Authors should clarify LMW. For example, is LWM less than 300? Or is a C5 diacid still LMW?

Line 41: LWM diacids -> Aerosols

Line 43: contribute from -> contribute to the wide range from

Line 56: is -> are

Line 62-64: This is a modeling study of cloud chemistry. There is no evidence yet from lab or field studies supporting this argument. There should be more discussions. Authors may discuss the mechanism of isoprene photooxidation focusing on partitioning of water soluble photooxidation products into the aqueous phase leading to oxalic acid formation.

Line 74: I don't see a verb in the sentence.

Line 89: I am not clear about this. What are the examples of chemical constraints? And what more need to be considered?

Line 101: identified -> observed

Line 175-178: C5 and C6 diacids are the ozonolysis products in smog chamber studies. How does that support photooxidation (OH radical reaction or photolysis) of anthropogenic precursors? If cloud (photo)chemistry was involved, authors need to state that. Besides, authors should provide evidence (or reference) of higher oxidation capacity in the areas (or polluted areas) during the summer than winter.

Line 188-200: OH reactions of water soluble organic compounds in the aqueous phase clearly produce malonic and succinic acid (Tan et al., EST, 2009). Authors should

include this in their discussion. How does aqueous photooxidation affect C3/C4?

Line 219-222: It seems this statement contradicts the previous statement (Line 184-186).

Line 221: Does "the low troposphere" means the ground level?

Line 232-233: Again, provide evidence or reference.

Line 245: I would note that C2 is oxalic acid. Let readers know that Cn is n-numbered carbon diacid somewhere in the text.

Line 248-250: Remove wC2 if you mean HCO-COOH is glyoxylic acid, which is an oxidation product of pyruvic acid, glyoxal, and methylglyoxal. Also reference Lim et al., ACP 2013, which show a full mechanism for aqueous OH reactions.

Line 252-255: I doubt this is true. Oxalic acid is the most dominant product for glyoxal with the high yield (Tan et al. EST, 2009; Lim et al., ACP, 2010). But it is not for methylglyoxal. So, the given correlation analysis cannot tell whether C2 is biogenic

Line 256-259: The same goes here. Is there any evidence support that winter is more anthropogenically influenced (e.g., seasonal emission inventory)?

Line 267-268: 2-methyltetrols are also isoprene SOA tracers. This means oxalic acid correlated with some of isoprene SOA tracers, not all.

Line 269: Are you sure there are diepoxy derivatives from isoprene? None of Paulot et al. Science, 2009 and Surratt et al. PNAS, 2010 reports that. If it is a typo, then change it to epoxy derivatives.

Line 270: further oxidation -> further gas-phase oxidation. There is no aqueous chemistry involved here.

Line 272-274: NOx is critical in isoprene-OH reactions. In the presence of NOx, methacrolein and MVK form. In the absence of NOx, ISOPOOH forms and further

ISOPOOH reactions produce IEPOX. To support your conclusion, you should provide NOx measurements.

Line 279-283: This statement is confusing. Are you saying oxalic acid is formed via gas-phase oxidation without aqueous reactions and lifted up to FT? But, there is no evidence of oxalic acid formation from gas-phase oxidation of isoprene or monoterpene. Besides, recent lab and field studies suggest that IEPOX contributes to SOA via aqueous chemistry (Nguyen et al. ACP, 2010; Budisulistiorini et al., ACP 2015; Pye et al., EST, 2013).

Line 281: What is the boundary layer height? Is this 2 km, the lower FT?

Line 283-285: It is difficult to conclude like this unless authors provide compelling evidence of aqueous chemistry leading to oxalic acid.

Line 291-299: These statements are also confusing. Why should oxalic acid be considered for SOA? Note that oxalic acid itself is not SOA due to high vapor pressure. I would agree if you argue that oxalic acid is evidence of aqueous chemistry. Then, you need to discuss how water soluble compounds formed from biogenic and anthropogenic sources; how they partition into wet aerosols or cloud waters; and how OH radicals formed and initiated aqueous-phase photooxidation. Besides, you mentioned C5 and C6 are from anthropogenic sources. But this chemistry is initiated by ozone and not related to aqueous chemistry. How is including this chemistry likely to reduce the discrepancy between model predictions and measurements of OA?

Line 308: Do you mean higher altitude at low FT?

Line 314: Specify "other species." Can they also provide evidence of aqueous chemistry like oxalic acid.

---

## Author Comment (AC1) · 6 May 2016

**Responses to Reviewer's Comments to**

Zhang et al., *"Aircraft observations of water-soluble dicarboxylic acids in the aerosols over China"*

Anonymous Referee #1

The authors present a summary of airborne filter measurement data with a focus on dicarboxylic acids in aerosols over China. The topic of organic acid composition in aerosols is of interest to the atmospheric chemistry community since the organic fraction of particles is complex and uncertain, and requires improvements to be able to model their impacts better. The paper is written fairly well but requires some minor English editing (I point out a few suggested changes below for improving the language).
The tables and figures are appropriate. The title and abstract are also appropriate in terms of representing the contents of the manuscript.
The methods used are fine, and the results are informative and rich. The strength of the work includes the detection of so many organic acids that exceed the range of acids detected in some past studies relying on airborne measurements. The overall conclusions mostly repeat those in other studies though, which I hope the authors can address in their revisions. This study confirms that organic acids are generated
by secondary production mechanisms and that aqueous processing can be important, especially aloft. To strengthen the paper, I would suggest that the authors try to find something new in their data to push the state of understanding of organic acids forward as compared to repeating what has already been documented. The data generated are of high value as airborne diacid data are scarce, so it would be useful to have this work published. But before that happens, as noted above, it is requested that the authors narrow in better on what is novel in their dataset to highlight better in their text. This may be assisted by better consideration of what past work has shown and how this dataset can extend upon those past papers, some of which are highlighted below.

**Response**: we thank the reviewer for the nice summary of our paper and the positive comments. In the following responses, we will reply to each comment listed below individually.

Major Comments:
The authors should expand discussion on the potential influence of biomass burning on their measurements. When was biomass burning evident and how did this impact the organic acid data? Presumably the authors have reasonable tracers for biomass burning plumes.

**Response**: In the revised MS, we discussed the effects of biomass burning on diacids as follows. "Levoglucosan (1,6-Anhydro-β-D-glucopyranose) has been used as a source tracer for biomass burning (Liu et al., 2013;Fu et al., 2012). Levoglucosan shows a significant correlation with oxalic acid and also secondary organic aerosol (SOA) tracers (Figure 5), indicating that that biomass burning is an important source of oxalic acid and SOA. It is interesting to note that levoglucosan was more abundant in spring than in summer. In addition, regression slope and correlation coefficient between oxalic acid and levoglucosan are higher in spring (i.e., slope: 2.7, r: 0.95) than in summer (i.e., slope: 1.7, r: 0.67), indicating that biomass-burning emissions play more important roles in spring than summer. Such higher values of slope of regression line and correlation coefficient were also found between levoglucosan and α/β-pinene- and β-caryophyllene-oxidation products, emphasizing an importance of springtime biomass burning. However, isoprene-oxidation products have a higher correlation coefficient with levoglucosan in summer than in spring, implying that biomass burning in summer is an important contributor of isoprene-derived SOA." Please see lines 338-350 in the revised MS.

[Figure]

**Figure R1**. Linear relationship for the concentrations of levoglucosan with oxalic acid, isoprene SOA tracers, α/β-pinene SOA tracers and β-caryophyllene SOA tracer from the aircraft measurements over China during (a) summer and (b) spring.

In various places in the manuscript the authors refer to Free Troposphere (e.g., Line 285). They should make it clear what data and criteria they used to distinguish between FT and the lower mixing layer.

**Response**: To clarify it, we change "at high altitudes within FT to "at high altitudes within the lower/middle troposphere". Similarly, we made changes in other places. Such terms have also been used in previous studies (Fu et al., 2014;Wang et al., 2007).

Since denuders are not mentioned in the instrument description, the authors should comment on what impact the lack of using denuders have on the data. The authors should make note of what affect temperature effects would have on their data as during aircraft sampling there likely are differences in temperature between ambient air and their filters.

**Response**: In the revised MS, we added the following sentences; "The lack of using organic denuders may lead to a positive artifact (e.g., 10% of the organic carbon) by possible adsorption of gas-phase organic acids on the quartz filters (Cheng et al., 2009), but this will not affect our conclusions. Because air conditioning was not available in the aircraft, the ambient temperatures inside and outside the cabin should be similar, and thus potential adsorption of gaseous organic acids on quartz filter should be minimal."

Specific Comments: Page 1, Line 4: change 'organics' to 'organic'

**Response**: Done.

Page 2, Line 38-41: Diacids have also been measured in other areas such as deserts and this should be noted for the sake of completeness (example provided here):
Sorooshian, A., et al. (2012). Hygroscopic and chemical properties of aerosols collected near a copper smelter: Implications for public and environmental health, Environ. Sci. Technol., 46, 9473-9480.

**Response**: We have referenced this paper.

Page 2, Line 41: 'play an important role in. . .'

**Response**: Changed.

Page 3, Line 66-69: For completeness the authors should refer to the following other airborne studies: For 'Coastal marine stratocumulus and cumulus clouds over USA', add:

Wonaschuetz, A., et al. (2012). Aerosol and gas re-distribution by shallow cumulus clouds: an investigation using airborne measurements, J. Geophys. Res., 117, D17202, doi:10.1029/2012JD018089.

Prabhakar, G., et al. (2014). Sources of nitrate in stratocumulus cloud water: Airborne measurements during the 2011 E-PEACE and 2013 NiCE studies, Atmos. Environ., 97, 166-173, doi:10.1016/j.atmosenv.2014.08.019.

Measurements have been conducted over inland agricultural and urban areas in the western United States: Sorooshian, A., et al. (2015). Surface and airborne measurements of organosulfur and methanesulfonate over the western United States and coastal areas, J. Geophys. Res., 120, doi:10.1002/2015JD023822.

**Response**: These papers are now cited.

Line 98: 'for a series. . .'

**Response**: Changed.

Line 325-327: The discussion here can benefit from inclusion of past work showing how the relative amount of oxalate (versus total organic mass) increases with relative humidity based on airborne measurements [Sorooshian, A., et al. (2010). Constraining the contribution of organic acids and AMS m/z 44 to the organic aerosol budget: On the importance of meteorology, aerosol hygroscopicity, and region, Geophys. Res. Lett.,37, L21807, doi:10.1029/2010GL044951] and how water-soluble organics have been shown to be enhanced relative to the surface when humid conditions aloft [Duong, H. T., et al. (2011). Water-soluble organic aerosol in the Los Angeles Basin and outflow regions: Airborne and ground measurements during the 2010 CalNex field campaign, J. Geophys. Res., 116, D00V04, doi:10.1029/2011JD016674.].

**Response**: We acknowledge the reviewer's comments. Although we did not measure relative humidity in these campaigns, the references were added in the revised MS.

Supplement Table 1: For altitude, the authors should include a plus/minus standard deviation since it seems unlikely that the aircraft was level for that long of a period for each sample.
**Response**: The accuracy in the altitude is roughly ±10%. This point has been added in the revised MS.

Line 356: What do the authors mean by the word 'control'? This seems like a very strong word that may not be warranted here. Are these diacids the majority of the OA mass and, if so, can this be shown in the manuscript in a revision?

**Response**: We change "control" to "have important impacts on".

Table 2: The authors should make mention of the sample number involved with these correlation calculations somewhere in the table or caption.

**Response**: Samples numbers for calculations are included in the revised caption.

**Anonymous Referee #2**

The manuscript by Zhang et al studied distributions of organic compounds obtained by aircraft measurements in China and argued that contribution of aqueous-phase photochemistry to SOA formation is significant using various correlation analyses. **Potentially this work will make an important contribution to SOA studies particularly in East Asia and be suitable for ACP readership.** But some of authors' arguments are confusing, misleading, and not clearly supported by evidence. Most of all, the organization is not effective. I highly recommend that authors should work on organization by bringing the LWC-oxalic acid analysis (currently in the end) to the front in Results and Discussion (Section 3) so that readers are clear about the main point of this manuscript is aqueous chemistry contribution to SOA. This will minimize confusions from paragraphs (Line 279-283, Line 283-285 and Line 291-299). Then, authors can discuss biogenic/anthropogenic influences on oxalic acid formation and aqueous chemistry. Provide biogenic/anthropogenic emission data if available. Provide NOx measurements, too for isoprene arguments. Take an advantage of referencing most recent papers. Below I provide comments. There are also numerous technical errors (e.g., grammatical errors, typos), which need to be corrected.

**Response**: We thank the reviewer for the nice summary of our paper and the positive comments. In the following we will respond to each comment listed below separately. In the revised MS, we moved the LWC-oxalic acid analysis (currently at the end) to the beginning of Results and Discussion section (Section 3). Further we included NOx data in the isoprene arguments, but we did not include biogenic/anthropogenic emission data because it is not available.

Line 38: Authors should clarify LMW. For example, is LWM less than 300? Or is a C5 diacid still LMW?

**Response**: In this paper, we measured diacids with carbon numbers from 2 to 11. To make it clearer, we remove "LMW" in the revised paper.

Line 41: LWM diacids -> Aerosols

**Response**: We changed the phrase to "As important components of aerosols, diacids …".
Line 43: contribute from -> contribute to the wide range from

**Response**: Changed.

Line 56: is -> are

**Response**: Changed.

Line 62-64: This is a modeling study of cloud chemistry. There is no evidence yet from lab or field studies supporting this argument. There should be more discussions. Authors may discuss the mechanism of isoprene photooxidation focusing on partitioning of water soluble photooxidation products into the aqueous phase leading to oxalic acid formation.

**Response**: We rephrased the sentence as "Indeed, from a model study it has been proposed that photochemical oxidation of isoprene and subsequent partitioning of water soluble photooxidation products into the aqueous phase is a predominant formation pathway of oxalic acid (Myriokefalitakis et al., 2011)."

Line 74: I don't see a verb in the sentence.

**Response**: We rephrased "maybe" to "may be".

Line 89: I am not clear about this. What are the examples of chemical constraints? And what more need to be considered?

**Response**: We changed "chemical constraints" to "chemical constraints (e.g., simultaneously measuring different types of organic aerosols such as diacids, ketoacids and α-dicarbonyls and SOA tracer compounds)".

Line 101: identified -> observed

**Response**: Changed.

Line 175-178: C5 and C6 diacids are the ozonolysis products in smog chamber studies. How does that support photooxidation (OH radical reaction or photolysis) of anthropogenic precursors? If cloud (photo)chemistry was involved, authors need to state that. Besides, authors should provide evidence (or reference) of higher oxidation capacity in the areas (or polluted areas) during the summer than winter.

**Response**: Although C5 and C6 diacids are the ozonolysis products in smog chamber studies, Pavuluri et al. (2015) recently proposed that C5 and C6 species are produced by photochemical processing of aqueous aerosols based on a laboratory experiment. In addition, C5 and C6 diacids have been used good tracers for anthropogenic sources in many studies (Kawamura and Bikkina, 2016). Using a global chemical transport model, the distributions of tropospheric OH over China show that tropospheric OH are higher in summer than in spring and winter (Shen and Wang, 2012;Su et al., 2012). These points have been added in the revised MS.

Line 188-200: OH reactions of water soluble organic compounds in the aqueous phase clearly produce malonic and succinic acid (Tan et al., EST, 2009). Authors should include this in their discussion. How does aqueous photooxidation affect C3/C4?

**Response**: In the revised MS, we added the following discussion, which reads as "In addition, C3 and C4 diacids can be formed by OH reactions of water soluble organic compounds such as glyoxal (Tan et al., 2009) and C3/C4 ratios gradually decrease with the reaction time (Pavuluri et al., 2015). The lower C3/C4 ratios in summer indicate that diacid aerosols are relatively fresh but will be subjected to more atmospheric aging than in other seasons due to increased OH concentrations and solar radiation." These points are added in the revised version.

Line 219-222: It seems this statement contradicts the previous statement (Line 184-186).

**Response**: This statement means that produced SOA is mostly fresh without intensive aging as evidenced by high M/F and low C3/C4 ratios. If SOA is intensively aged, M and C4 will be transformed to F or degraded to C3, respectively, but it is not observed in our study.

Line 221: Does "the low troposphere" means the ground level?

**Response**: We rephrased "the low troposphere" to "the lower/middle troposphere", to be consistent with previous terms used in the same campaign (Fu et al., 2014;Wang et al., 2007).

Line 232-233: Again, provide evidence or reference.

**Response**: References are added (Shen and Wang, 2012;Su et al., 2012).

Line 245: I would note that C2 is oxalic acid. Let readers know that Cn is n-numbered carbon diacid somewhere in the text.

**Response**: Following the comment, we rephrased the sentence.

Line 248-250: Remove wC2 if you mean HCO-COOH is glyoxylic acid, which is an oxidation product of pyruvic acid, glyoxal, and methylglyoxal. Also reference Lim et al., ACP 2013, which show a full mechanism for aqueous OH reactions.

**Response**: We revised the MS by taking the comment.

Line 252-255: I doubt this is true. Oxalic acid is the most dominant product for glyoxal with the high yield (Tan et al. EST, 2009; Lim et al., ACP, 2010). But it is not for methylglyoxal. So, the given correlation analysis cannot tell whether C2 is biogenic.

**Response**: Yes, we agree with the comment. We change the sentences as "We found that the correlation coefficient between Gly and $C_2$ is higher than that for MeGly and $C_2$ in winter and spring samples, being consistent with the fact that oxalic acid is the dominant product of glyoxal with the high yield but not for methylglyoxal (Tan et al., 2009; Lim et al., 2010)."

Line 256-259: The same goes here. Is there any evidence support that winter is more anthropogenically influenced (e.g., seasonal emission inventory)?

**Response**: We do not have emission inventory data, but a recent study reveals that air pollutants such as $SO_2$, $NO_2$, PM2.5 and PM10 are higher in winter than in summer and these pollutants are mostly from anthropogenic emissions (Zhang and Cao, 2015). This is now included.

Line 267-268: 2-methyltetrols are also isoprene SOA tracers. This means oxalic acid correlated with some of isoprene SOA tracers, not all.

**Response**: we rephrased the sentence as "Oxalic acid shows a strong positive correlation with *two specific* isoprene-derived SOA tracers such as 2-methylglyceric acid (2-MGA) and $C_5$-alkene triols (cis-2-methyl-1,3,4-trihydroxy-1-butene,3-methyl-2,3,4-trihydroxy-1-butene plus trans-2-methyl-1,3,4-trihydroxy-1-butene)."

Line 269: Are you sure there are diepoxy derivatives from isoprene? None of Paulot et al. Science, 2009 and Surratt et al. PNAS, 2010 reports that. If it is a typo, then change it to epoxy derivatives.

**Response**: It should be epoxy derivatives. We rephrased it as epoxy derivatives.

Line 270: further oxidation -> further gas-phase oxidation. There is no aqueous chemistry involved here.

**Response**: Corrected.

Line 272-274: NOx is critical in isoprene-OH reactions. In the presence of NOx, methacrolein and MVK form. In the absence of NOx, ISOPOOH forms and further ISOPOOH reactions produce IEPOX. To support your conclusion, you should provide NOx measurements.

**Response**: Overall, tropospheric atmosphere over China was characterized with relative high NOx concentrations. We rephrased the sentence as "Such a good correlation between 2-MGA and oxalic acid could demonstrate that oxalic acid has a very close link with the higher-generation products of isoprene in the presence of relatively high $NO_x$ (i.e., $NO_x$ averaged to 6.6±4.0 ppbv in summer and 3.9±3.3 ppbv), which could serve as precursors of oxalic acid over China."

Line 279-283: This statement is confusing. Are you saying oxalic acid is formed via gas-phase oxidation without aqueous reactions and lifted up to FT? But, there is no evidence of oxalic acid formation from gas-phase oxidation of isoprene or monoterpene. Besides, recent lab and field studies suggest that IEPOX contributes to SOA via aqueous chemistry (Nguyen et al. ACP, 2010; Budisulistiorini et al., ACP 2015; Pye et al., EST, 2013).
**Response**: We modified the sentence as "Based on the consistent good correlations of oxalic acid with SOA tracers derived from isoprene, monoterpene and β-caryophyllene, we propose that a large fraction of oxalic acid in the lower/middle troposphere over China is of secondary origin, i.e., via aqueous chemistry."

Line 281: What is the boundary layer height? Is this 2 km, the lower FT?

**Response**: The boundary layer height was not measured. We rephrased "the boundary layer height" to "the lower/middle troposphere" to be consistent with terms used in this and previous study.

Line 283-285: It is difficult to conclude like this unless authors provide compelling evidence of aqueous chemistry leading to oxalic acid.

**Response**: Based on the consistent good correlations of oxalic acid with specific SOA tracers in aerosols, we could propose that oxalic acid is of secondary origin. We rephrased "should" to "may".

Line 291-299: These statements are also confusing. Why should oxalic acid be considered for SOA? Note that oxalic acid itself is not SOA due to high vapor pressure. I would agree if you argue that oxalic acid is evidence of aqueous chemistry. Then, you need to discuss how water soluble compounds formed from biogenic and anthropogenic sources; how they partition into wet aerosols or cloud waters; and how OH radicals formed and initiated aqueous-phase photooxidation.

**Response**: Although oxalic acid has relatively high vapor pressure, it has been observed as the most abundant individual organic compounds in aerosols (Kawamura and Bikkina, 2016). Our study together with many previous studies have revealed that oxalic acid is mostly from aqueous-phase production and is therefore of secondary origin (Myriokefalitakis et al., 2011). In addition, oxalic acid may partly exist as hydrated and/or salt forms in aerosols. Therefore, it is important to consider oxalic acid or oxalate as important SOA tracers. Please see lines 332-337 in the revised MS.

Besides, you mentioned C5 and C6 are from anthropogenic sources. But this chemistry is initiated by ozone and not related to aqueous chemistry. How is including this chemistry likely to reduce the discrepancy between model predictions and measurements of OA?

**Response**: We removed the following sentences "~~Inclusion of oxalic acid (and also other LWM diacids) as major products from atmospheric oxidation of biogenic (and also anthropogenic) VOCs may partially reduce the discrepancy between modeled and observed tropospheric OA during the ACE-Asia campaign , although further studies are still required to investigate other SOA compounds (e.g., oligomeric components) produced from anthropogenic and biogenic VOCs in the reactions to fully understand the associated formation pathway and mechanism~~".

Line 308: Do you mean higher altitude at low FT?

**Response**: We rephrased it as "in the upper troposphere (~2 km)."

Line 314: Specify "other species." Can they also provide evidence of aqueous chemistry like oxalic acid.

**Response**: This sentence has been removed in the revised MS.

**References:**

[revised manuscript text omitted]